

# Towards a quantum fluid theory of correlated many-fermion systems from first principles

Z. A. Moldabekov[1,2], T. Dornheim[1,2], G. Gregori[3],
F. Graziani[4], M. Bonitz[5] and A. Cangi[1,2*]

**1** Center for Advanced Systems Understanding (CASUS), D-02826 Görlitz, Germany
**2** Helmholtz-Zentrum Dresden-Rossendorf, D-01328 Dresden, Germany
**3** Department of Physics, University of Oxford, Parks Road, Oxford OX1 3PU, UK
**4** Lawrence Livermore National Laboratory, Livermore, CA, 94550, USA
**5** Institut für Theoretische Physik und Astrophysik, Christian-Albrechts-Universität zu Kiel,
Leibnizstraße 15, 24098 Kiel, Germany

* a.cangi@hzdr.de

## Abstract

Correlated many-fermion systems emerge in a broad range of phenomena in warm dense matter, plasmonics, and ultracold atoms. Quantum hydrodynamics (QHD) complements first-principles methods for many-fermion systems at larger scales. We illustrate the failure of the standard Bohm potential central to QHD for strong perturbations when the density perturbation is larger than about $10^{-3}$ of the mean density. We then extend QHD to this regime via the *many-fermion Bohm potential* from first-principles. This may lead to more accurate QHD simulations beyond their common application domain in the presence of strong perturbations at scales unattainable with first-principles methods.



# 1  Introduction

Correlated quantum many-fermion systems are currently in the focus of several fields ranging from high-energy-density physics [1] to ultracold fermionic atoms [2] and correlated materials [3]. Progress in all these fields relies on accurate theory and simulations including quantum Monte Carlo (QMC) [4], density functional theory (DFT) [5,6], nonequilibrium Green functions [7], and density matrix renormalization group (DMRG) methods [8]. While remarkable progress was achieved with these methods, their high computational cost and fundamental bottlenecks significantly restrict their application. For example, the fermion sign problem complicates the use of QMC [9], or the computational cost renders DMRG applications in three spatial dimensions infeasible. Therefore, there is a high need for complementary methods that extend the domain of simulations to length and time scales relevant for experiments, even at the price of reduced accuracy.

One such method is quantum hydrodynamics (QHD). There has recently been a surge of activities based on QHD in a number of research areas including warm dense matter (WDM) [10–13], plasmonics [14–16], electron transport in semiconductor devices and thin metal films [17, 18], reactive scattering [19,20], cosmology, and dark matter research [21–23].

QHD complements the aforementioned first-principles methods by enabling simulations at larger length and longer time scales. The quantum Bohm potential is central to QHD [13,17]. It captures quantum tunneling, spill out, and other non-local effects. Commonly, the quantum Bohm potential is approximated as [17,24,25]

$$v_B(\mathbf{r}, t) = -\hbar^2/(2m)\left[\nabla^2\sqrt{n(\mathbf{r}, t)}/\sqrt{n(\mathbf{r}, t)}\right] \tag{1}$$

in terms of the mean density of electrons $n(\mathbf{r}, t)$; hereafter called *standard Bohm potential*. It is utilized in this form to model phenomena in various many-fermion systems.

While standard QHD has proven useful, we question the validity of the standard Bohm potential when strong density perturbations are present. These emerge, for example, in strongly perturbed WDM [26] and quantum plasmas [27, 28]. Most notably, this regime is probed in recent and upcoming X-Ray scattering measurements of matter that is shock-compressed and laser-excited [29–34] using the seeding technique discussed in the conclusions.

In this research report, we therefore extend QHD to the regime of strong density perturbations. Our central result is to utilize the *many-fermion Bohm potential* [11]

$$\tilde{v}_B(\mathbf{r}, t) = -\frac{\hbar^2}{2mN}\sum_{i=1}^{N} f_i \frac{\nabla^2\sqrt{n_i(\mathbf{r}, t)}}{\sqrt{n_i(\mathbf{r}, t)}}, \tag{2}$$

where $N$ is the total number of electrons, $n_i = |\phi_i|^2$ represents the amplitude of an orbital with the occupation number given by the Fermi function $f_i(\beta, \mu) = [\exp\{\beta(\epsilon_i - \mu)\} + 1]^{-1}$ at finite temperature $k_B T = \beta^{-1}$. Specifically, we (1) generate an exact *many-fermion Bohm potential* based on *exact* QMC data, (2) show how the standard Bohm potential differs significantly from the *many-fermion Bohm potential* for strong density perturbations, and (3) highlight how the resulting forces – the key ingredient to QHD – differ greatly in this regime. Throughout the manuscript, we consider the practically important example of the harmonically perturbed, interacting electron gas at finite temperature which is a challenging many-fermion system and is a relevant for modeling high-energy density experiments conducted at coherent light sources and pulsed power facilities around the globe.

Utilizing the *many-fermion Bohm potential* in QHD is motivated by the fact that it is derived from the exact quantum dynamics of electrons within time-dependent DFT [6] which provides the crucial link between QHD and interacting many-fermion systems.

## 2 Theory

We begin with the non-relativistic, many-particle Hamiltonian of interacting fermions

$$\hat{H} = \hat{T} + \hat{V}_{ee} + \hat{V}, \tag{3}$$

where $\hat{T}$ denotes the kinetic energy operator, $\hat{V}_{ee}$ the electron-electron interaction and $\hat{V}$ the external potential including the ionic background. The solutions are $N$-particle wave functions that are antisymmetric and normalized. For the sake of clarity we consider only spin-unpolarized systems. A formally exact and computationally feasible solution to the quantum dynamics of electrons is given within time-dependent DFT [6]. Here, a set of $N$ time-dependent Kohn-Sham (KS) equations

$$i\hbar\frac{\partial}{\partial t}\phi_i(\mathbf{r},t) = \left[-\frac{\hbar^2}{2m}\nabla^2 + v_{\mathrm{s}}(\mathbf{r},t)\right]\phi_i(\mathbf{r},t), \tag{4}$$

yields the exact time evolution of the electronic density, $n(\mathbf{r},t) = \sum_i f_i|\phi_i(\mathbf{r},t)|^2$, in terms of the single-particle KS orbitals $\phi_i(\mathbf{r},t)$, where $f_i$ denotes an occupation function. This is achieved by the KS potential, $v_{\mathrm{s}}(\mathbf{r},t) = v(\mathbf{r},t) + v_{\mathrm{H}}[n](\mathbf{r},t) + v_{\mathrm{xc}}[n](\mathbf{r},t)$, which exactly mimicks the electron-electron interaction within a mean-field description. Here, $v$ denotes the external potential, $v_{\mathrm{H}}[n]$ the classical electrostatic (Hartree) potential, and $v_{\mathrm{xc}}[n]$ the exchange-correlation potential.

Now, the time-dependent KS equations are reformulated into a set of coupled QHD equations by the following steps: (1) we insert the amplitude-phase representation of the KS orbitals [35], $\phi_i(\mathbf{r},t) = \sqrt{n_i(\mathbf{r},t)}\exp[iS_i(\mathbf{r},t)]$, into the time-dependent KS equations; (2) we use the expression for the mean orbital density, $\bar{n}(\mathbf{r},t) = \sum_i f_i n_i(\mathbf{r},t)/N$, and velocity, $\mathbf{v} = \sum_i f_i\mathbf{v}_i/N$, where we introduce $n_i(\mathbf{r},t) = |\phi_i(\mathbf{r},t)|^2$, as the KS orbital density and $\mathbf{v}_i = \nabla S_i(\mathbf{r},t)/m$, as the KS orbital velocity; (3) we introduce density and velocity fluctuations $n_i = \bar{n} + \delta n_i$ and $\mathbf{v}_i = \mathbf{v} + \delta\mathbf{v}_i$. These steps yield the formally exact QHD equations

$$\frac{\partial\bar{n}}{\partial t} + \frac{1}{N}\sum_i f_i\nabla\cdot(n_i\mathbf{v}_i) = 0, \tag{5}$$

$$m\frac{\partial\mathbf{v}}{\partial t} = -\nabla\tilde{v}_B - \frac{1}{n}\nabla P_e + \frac{1}{n}\nabla\cdot\boldsymbol{\sigma}_e + e\mathbf{E} - \nabla v_{\mathrm{xc}}, \tag{6}$$

where we have not yet made any assumptions about velocity and density fluctuations [11]. In Eq. (6), $e$ is the absolute value of the electron charge, $P_e = \frac{1}{2m}\partial_\alpha\overline{\delta p_{i\alpha}^2}$ the electronic pressure term (with $\delta\mathbf{p}_i = m\delta\mathbf{v}_i$), $\boldsymbol{\sigma}_e = \frac{1}{m}\partial_\gamma\overline{\delta p_{i\alpha}\delta p_{i\gamma}}$ with $\gamma \neq \alpha$ the electronic viscous stress-tensor, and $\mathbf{E} = -\nabla[v + v_{\mathrm{H}}]$ the electric field due to the Hartree and external potentials. The first equation is the continuity equation, whereas the second is the momentum conservation equation. Notice that the *many-fermion Bohm potential* emerges naturally [11]. These QHD equations are equivalent to the time-dependent KS equations.

The QHD equations are turned into computationally feasible practice by employing approximations to (1) the exchange-correlation functional $v_{\mathrm{xc}}$, (2) the equation of state $P_e$, (3) the viscous stress-tensor $\boldsymbol{\sigma}_e$, and (4) setting $\frac{1}{N}\sum_i f_i\nabla\cdot(n_i\mathbf{v}_i) = \nabla\cdot(\bar{n}\mathbf{v})$ in Eq. (5) where the averaged fluctuations of a flux $\langle\delta\bar{\mathbf{j}}\rangle = \langle\delta n_i\delta\mathbf{v}_i\rangle$ are assumed to be negligible compared to the mean value $\bar{\mathbf{j}} = \bar{n}\mathbf{v}$. Proven approximations to the exchange-correlation energy, i.e., $v_{\mathrm{xc}}$ can be employed where recent developments such as the parametrization of the interacting electron gas at finite temperature [36] provide a solid basis for an accurate inclusion of exchange-correlation effects into the QHD equations. Using approximations to the equation of state and the viscous stress-tensor enables QHD to go beyond the length and time scales

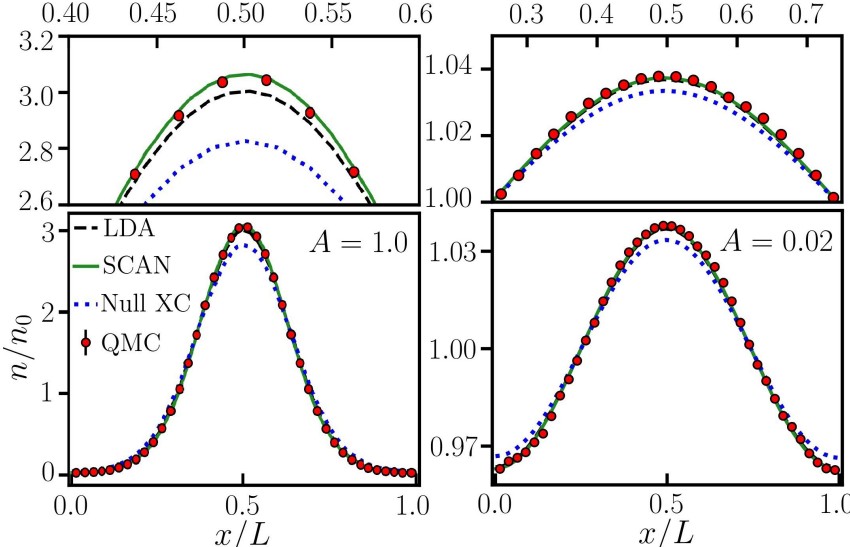

Figure 1: Electronic density for two different amplitudes $A$, at $r_s = 2$ and $\theta = 1$. QMC results (red circles) are compared to KS-DFT data for different XC-%potentials: solid green: SCAN; dashed black: LDA; dotted blue: non-interacting fermions ($v_{\mathrm{xc}} = 0$).

that are attainable in time-dependent DFT calculations. Commonly, the electronic pressure $P_e$ is approximated by the ideal Fermi gas. On the other hand, practical calculations spanning a large range of length and time scales are performed with classical hydrodynamics simulations. These, however, completely neglect quantum non-locality effects. As discussed below, these quantum effects become increasingly relevant for high-energy-density sciences due to ongoing and recent developments in experimental and diagnostic capabilities. We stress that in traditional QHD used in prior works, the many-fermion Bohm potential is approximated by the standard Bohm potential in terms of the mean density as defined in Eq. (1).

## 3 Results

As the central result of this work we demonstrate the relevance of the *many-fermion Bohm potential* for the QHD equations (5) and (6), whereas in all prior works the standard Bohm potential was used. First, we generate a *many-fermion Bohm potential* using KS-DFT based on exact QMC calculations of the harmonically perturbed, interacting electron gas. Then we show that the standard Bohm potential differs both qualitatively and quantitatively from $\tilde{v}_B$ to a great extent for strong density perturbations. Finally, we illustrate how these deviations yield vastly different forces. We, hence, argue that these lead to a different quantum plasma dynamics when used in the QHD equations. Agreement to better than 50% in the resulting forces is achieved only for small density perturbations when $|\delta n| \lesssim 10^{-3} n_0$ or $q > 2 q_F$. This is further analyzed in Appendix C. The use of the *many-fermion Bohm potential* now renders QHD valid for the regime of strong density perturbations. While approximations to the pressure and viscous stress-tensor also influence the accuracy of the QHD equations, we focus on the *many-fermion Bohm potential*. It primarily determines the accurate inclusion of quantum effects, e.g., tunneling and spill-out, that are crucial for the aforementioned applications.

An important application that is highly relevant for high-energy-density physics is the har-

monically perturbed, interacting electrons gas. It is described by the Hamiltonian [37–39]

$$\hat{H} = \hat{H}_{\text{UEG}} + \sum_{i=1}^{N} 2A\cos\left(\boldsymbol{r}_i \cdot \boldsymbol{q}\right),\tag{7}$$

where $\hat{H}_{\text{UEG}}$ denotes the Hamiltonian of the uniform electron gas with periodic boundary conditions. We choose the x-axis along **q** with $q = jq_{\min}$, $q_{\min} = 2\pi/L$, $L = (N/n_0)^{1/3}$, and $n_0$ the number density of electrons.

The electronic states described by Eq. (7) are generated in recent WDM experiments (see Conclusions 4 for further details). The amplitude $A$ in Eq. (7) controls the character of the KS orbitals. Tuning $A$ changes the KS orbitals from plane waves to strongly localized wave functions. Moreover, by varying both $A$ and the wave number $q$, we tune the density gradients from small to large. The relevant parameter space is spanned by the density parameter $r_s = a/a_B$ and the degeneracy parameter $\theta = k_B T/E_F$, where $a$ is the mean inter-electronic distance, $a_B$ the first Bohr radius, $T$ the temperature, and $E_F$ the Fermi energy. For the remainder of this paper we choose $r_s = 2$ and $\theta = 1$. This corresponds to the WDM and quantum plasma regime [1, 4].

The construction of the *many-fermion Bohm potential* relies on accurate KS orbitals. We generate orbitals using KS-DFT for various amplitudes $10^{-3} \leq A \leq 1$ corresponding to the range from weak to strong perturbations. We assess their accuracy by comparing them with the exact result provided by QMC calculations. Details of the KS-DFT and QMC calculations are provided in Appendix A and Appendix B, respectively. The electronic densities for $A = 1$ and $A = 0.02$, using various exchange-correlation approximations (non-interacting fermions, LDA [40], and SCAN [41]) are illustrated in Fig. 1, where $q = q_{\min} = 0.84q_F$. The comparison with the QMC data (red circles) confirms that the KS-DFT calculations using the SCAN functional provide the KS orbitals that virtually yield the exact density.

We now construct an exact *many-fermion Bohm potential* by inserting these KS orbitals into Eq. (2). The results are shown in the top panel of Fig. 2 for $q = 0.84q_F$. They are ordered in increasing perturbation strength ($A = 0.02, 0.1, 0.5$). At the top we compare the *many-fermion Bohm potential* (thick green) with the standard Bohm potential (dashed blue). We observe significant differences for all amplitudes, and profound qualitative differences at high perturbation strength ($A = 0.5$). To better understand the origin of these differences, consider contributions of the individual KS orbitals with a maximum and a minimum in the central region (orange and red lines). The former lead to a stronger *many-fermion Bohm potential* in the density depletion region at the edges, whereas the latter yield a weaker *many-fermion Bohm potential* in the central region where electrons accumulate. The important point to note is that the contribution from individual orbitals does not depend on the amplitude of the orbital density, but on its shape as is apparent from Eq. (2). This means that the contribution of a highly curved orbital can be critical, even if the corresponding occupation number may be relatively small.

Next, we relate these differences to the relevant energy scale in the QHD equations. We compare against the ideal part of the free energy density, $f_{\text{TF}}[n(\mathbf{r})] = \delta F_{\text{id}}[n]/\delta n(\mathbf{r}) = \mu$ (red squares), which is a common approximation to the pressure in the QHD equations [13, 17] in terms of the Thomas-Fermi (TF) free-energy functional. The top panel of Fig. 2 shows that the ideal part of the free energy density has about the same order of magnitude as the standard Bohm potential throughout highlighting the importance of the *many-fermion Bohm potential*.

Now we assess the impact of using the *many-fermion Bohm potential*, instead of the standard Bohm potential, for simulating quantum dynamics. We compute the force due to the pressure of the quantum Bohm potential from $n(\boldsymbol{r})\nabla V_B$, where $V_B$ is the either the standard Bohm potential $v_B$ or the *many-fermion Bohm potential* $\tilde{v}_B$. To assess the importance of the observed differences, we compare them with the force due to TF pressure, $\nabla P_{\text{TF}} = n(\boldsymbol{r})\nabla f_{\text{TF}}[n(\boldsymbol{r})]$.

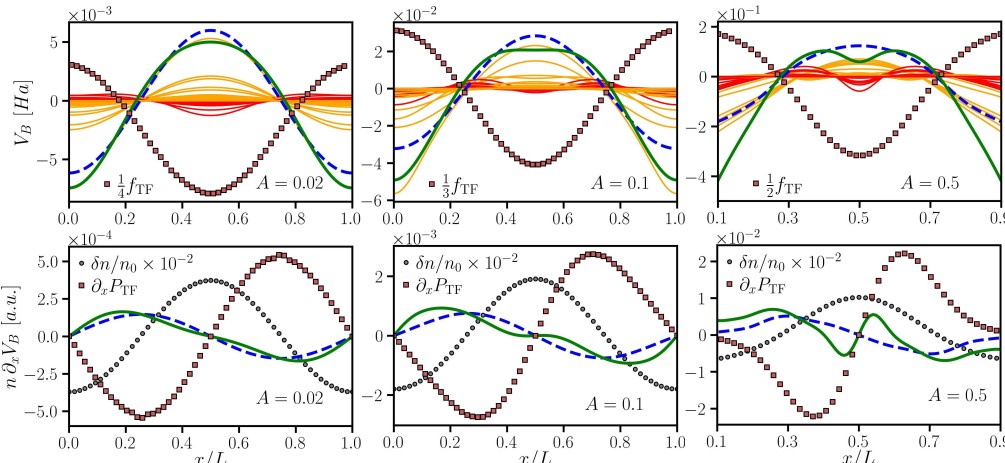

Figure 2: Upper panel: Comparison of the exact *many-fermion Bohm potential* (thick green) with the standard Bohm potential (dashed blue) at $r_s = 2$ and $\%\theta = 1$. Additionally, the TF free energy density (red squares, scaled) and the contributing KS orbitals are illustrated (thin red (dark) and orange (light) lines for the contributions with the local maximum and minimum in the central region, respectively). The contribution of orbitals is scaled by a factor two (three) at $A = 0.02$ ($A = 0.1$ and $A = 0.5$). Lower panel: Comparison of the forces from the *many-fermion Bohm potential* (green) with forces from the standard Bohm potential (dashed blue). We also display the TF pressure (squares) and the density profile (grey circles). Note the scaling.

The lower panel of Fig. 2 demonstrates that the forces differ distinctly. At small perturbation strength the maximum deviation of the forces is already 50%. This deviation further increases with a stronger perturbation amplitude. At $A = 0.5$, they differ substantially, and the standard Bohm potential fails to even yield a qualitative description. In the central region they are also qualitatively very different. Furthermore, the comparison with the TF force highlights the relative importance of the force due to $\tilde{v}_B$. At $A = 0.02$, the TF force is about four times stronger than the force due to both variants of the quantum Bohm potential. With increasing perturbation strength, the force due to the *many-fermion Bohm potential* becomes more relevant. At $A = 0.5$, it is close to the TF force in the central region, whereas it even exceeds the TF force in the density depletion regions close to the edges.

Next, in Fig. 3, we provide a more detailed comparison of the forces. On the left, we show the ratio of the forces due to the *many-fermion Bohm potential* and the standard Bohm potential, whereas on the right, we show the ratio of the force due to $\tilde{v}_B$ with the TF force at $A = 0.1, 0.3, 1.0$. We infer that, in general, the force due to the standard Bohm potential differs from the result based on the exact *many-fermion Bohm potential* by at least a factor of two throughout (left panel). For a small perturbation amplitude, $A = 0.1$, the standard Bohm potential based result significantly overestimates (up to fifty times) the exact *many-fermion Bohm potential* based data in the central region and underestimates it by a factor of two in the density depletion region. At larger amplitudes ($A = 0.3$ and $A = 1.0$), the differences in both the density depletion region and in the central region increase. Finally, in Fig. 3 (right), we assess the relative importance of the quantum Bohm potentials. We deduce that the force due to the *many-fermion Bohm potential* is dominant in the density depletion regions, when $A \gtrsim 0.3$, with a maximum value of the density increase of $|\delta n| \gtrsim 0.6 \, n_0$. In conclusion, the *many-fermion Bohm potential* may lead to a substantially different quantum dynamics which will be explored in our future work.

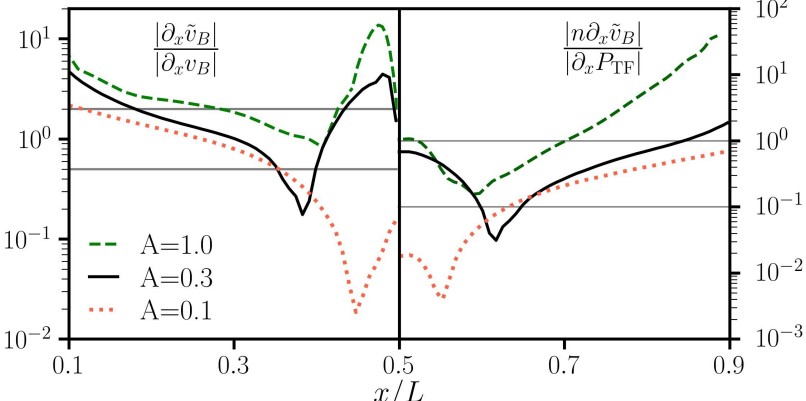

Figure 3: Left: Ratio of the forces between the exact *many-fermion Bohm potential* ($\tilde{v}_B$) and the standard Bohm potential ($v_B$) at $r_s = 2$ and $\theta = 1$ for increasing density perturbation amplitudes $A$. Right: Ratio of the forces due to the *many-fermion Bohm potential*, $\tilde{v}_B$, and the TF pressure.

## 4 Conclusions and Outlook

For a degenerate quantum many-particle system with Bose statistics in the condensate, the Madelung decomposition leads to the Gross-Pitaevski equation. Here, the exactness of the standard Bohm potential can be proven [11]. For fermionic systems, such a proof does not exist. A trivial exception is the case when the amplitudes of all orbitals coincide and the system is mapped onto a single orbital [11].

In this work, we carried out the very first investigation of the *many-fermion Bohm potential* for a correlated many-fermion system based on first-principles data from QMC and KS-DFT. Despite its long history in quantum mechanics since its derivation by Bohm in 1952 [35] and its importance as a computational device in QHD, this has not been attempted before. Our key result highlights the very limited applicability of the standard Bohm potential which is used in virtually all previous works of QHD. Considering a harmonic perturbation in the Hamiltonian defined in Eq. (7), we showed that the standard Bohm potential is only valid for a very weakly perturbed electron gas ($|\delta n| \lesssim 10^{-3} n_0$) or at very large wave-numbers ($q > 2q_F$). Likewise, we demonstrated that the *many-fermion Bohm potential* is needed to model nonlinear phenomena in quantum plasmas and WDM. We further illustrated the significance of the force produced by the *many-fermion Bohm potential* for QHD simulations.

We anticipate that taking into account the *many-fermion Bohm potential* in quantum fluid approaches will play a significant role for many upcoming high-energy-density physics experiments. Strongly perturbed WDM states are generated and probed, for example, using THz lasers with an intensity of 600 kV/cm that corresponds to a perturbation amplitude of $A \simeq 0.3$ [29] and using free electron lasers with intensities of up to $10^{22}$ W/cm$^2$ that lead to $A \approx 2$ [27]. Likewise, it was recently demonstrated in an experiment [42] that spatially modulated WDM is created by laser pumping of a sample with a pre-designed, periodic grating structure. The induced WDM states can be characterized in-situ with the small-angle x-ray scattering technique using femtosecond X-Ray free-electron laser pulses on a spatial resolution of nanometers.

Another exciting application of QHD is inertial confinement fusion [43] where strongly inhomogeneous electronic states emerge in the heating of shock-compressed fuel capsules. Of particular interest is the effect the *many-fermion Bohm potential* has on the shock behavior in high-energy density applications using lasers or pulsed power. The presence of higher-order

spatial derivatives of the density produces a dissipative-like effect on the shock structure, shearing the interface and broadening the shock front. Particularly, the effect of the standard Bohm potential has very recently been assessed in hydrodynamics simulations [44]. As demonstrated in Figs. 2 and 3, the *many-fermion Bohm potential* yields significantly different forces in regions of strong density perturbation than the standard Bohm potential. We therefore expect the *many-fermion Bohm potential* to further impact the dynamics of shock formation.

Other interesting applications include non-linear wave phenomena and instabilities in quantum plasmas [13]. We also expect the *many-fermion Bohm potential* to impact the field of nano-plasmonics [14–16] where simulations of large nano-clusters are routinely performed with QHD. Moreover, the *many-fermion Bohm potential* might enable quantum dynamics simulations of cold atom experiments that study transport properties [2]. We also speculate that the force field generated by the *many-fermion Bohm potential* can be utilized as a computationally inexpensive neural-network surrogate model as it was done, e.g., for the free energy functional in KS-DFT [45, 46] and the local field correction in QMC [47].

Finally, the *many-fermion Bohm potential* awaits exciting applications in cosmology. These approaches are based on an observation made by de Broglie pointing out that quantum mechanical effects are entirely equivalent to a conformal transformation of the background metric [48, 49]. This leads to a representation of the non-local Bohm potential of all the particles in the Universe as an effective cosmological constant [21]. Therefore, this outlines an interesting line of future research.

# Acknowledgements

**Author contributions**   Z. A. Moldabekov performed the KS-DFT simulations, analyzed the results, contributed to derivations in the theory section and made a major contribution to writing the paper. T. Dornheim performed QMC simulations and contributed to writing the paper. G. Gregori contributed to writing the paper and provided insights on relevance of the results for cosmology. F. Graziani contributed to writing the paper and provided insights on the relevance of the results for hydrodynamics in high-energy-density plasmas. M. Bonitz contributed to derivation of the many-fermion Bohm potential and to writing the paper. A. Cangi made major contributions to writing the paper and contributed to derivations in the theory section.

**Funding information**   This work was partly funded by the Center for Advanced Systems Understanding (CASUS) which is financed by Germany's Federal Ministry of Education and Research (BMBF) and by the Saxon Ministry for Science, Culture and Tourism (SMWK) with tax funds on the basis of the budget approved by the Saxon State Parliament. We gratefully acknowledge CPU-time at the Norddeutscher Verbund für Hoch- und Höchstleistungsrechnen (HLRN) under grant shp00026 and on a Bull Cluster at the Center for Information Services and High Performance Computing (ZIH) at Technische Universität Dresden. The work of GG was funded in parts by the Engineering and Physical Sciences Research Council (grant numbers EP/M022331/1 and EP/N014472/1). MB acknowledges support from the Deutsche Forschungsgemeinschaft via grant BO1366/15. A part of this work was performed under the auspices of the US Department of Energy by Lawrence Livermore National Laboratory under Contract DE-AC5207NA27344.

# A KS-DFT simulation details

The KS-DFT calculations were performed with GPAW [50], which is a real-space implementation of the projector augmented-wave method. A $k$-point grid of $12 \times 12 \times 12$ using Monkhorst-Pack sampling of the Brillouin zone (**k**-points) was used. At $\theta = 1$, 180 orbitals (with the smallest occupation number of about $10^{-4}$) were used for a total of 14 electrons. The grid spacing was set to 0.15 for $10^{-3} \le A \le 1$ and $0.3\,q_F \lesssim q \lesssim 2.53\,q_F$. The Hamiltonian of electrons is given by the sum of the standard (unperturbed) uniform electron gas Hamiltonian and the potential energy term corresponding to external perturbation. Several exchange-correlation (XC) functionals were used: the standard LDA functional by Perdew-Zunger for degenerate electrons [51], the GDSMFB functional which is a parametrization of the LDA of the homogeneous electron gas at finite temperature [52], PBE [53], PBEsol [54], AM05 [55], and the meta-GGA functional SCAN [41]. The ab-initio quality of the KS-DFT calculations was validated with first-principles quantum Monte Carlo (QMC) calculations. We found that SCAN reproduces the exact QMC data more accurately than any of the other XC functionals. Only in the limit of a weak perturbation, the tested XC functionals yield agreement, as they reduce to the zero-temperature limit of the LDA.

The relative error of the results obtained using different XC functionals compared to the QMC data is given in Fig. 4 for $A = 1$ and $A = 0.02$. The corresponding total density is presented Fig. 1 of the manuscript. The case $A = 1$ corresponds to a strong-perturbation regime with a minimum density of $n \simeq 0.03\,n_0$ close to the edges of the simulation box and with a maximum density of $n \simeq 3\,n_0$ in the center. The case $A = 0.02$ corresponds to the weak-perturbation regime with a density maximum $n \simeq 1.04\,n_0$ and minimum $n \simeq 0.96\,n_0$. From Fig. 4 we infer that the inclusion of XC effects by using the LDA significantly improves upon the fully non-interacting case (no XC functional) with a maximum error of about 4.7% in the central region for $A = 1$. However, using GGA functionals does not improve over the LDA results. The exact QMC data are reproduced remarkably well by the SCAN functional with an accuracy better than 1.43 %. We conclude that it is crucial to go beyond both LDA and GGA in order to obtain an accurate density when the perturbation is strong. A further analysis of this observation is presented in Ref. [56].

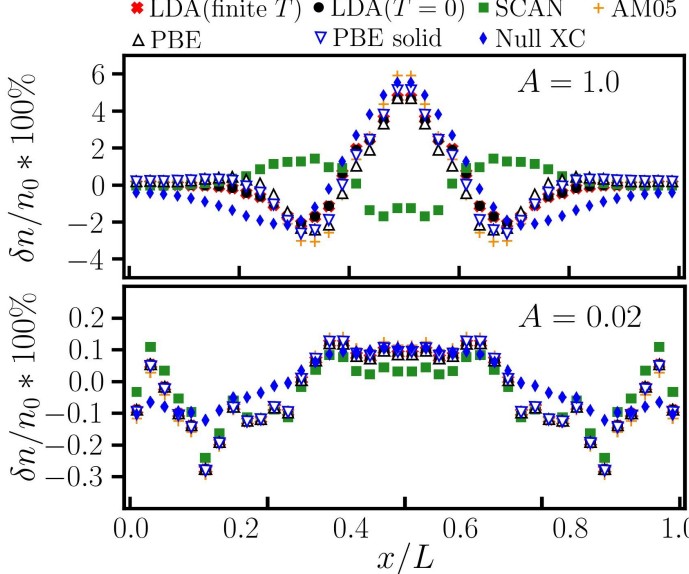

Figure 4: Relative error of the density using different XC functionals compared to the QMC data at $\theta = 1$ and $r_s = 2$.

## B   QMC simulation details

We use the standard path integral Monte Carlo (PIMC) method [4] without any nodal restrictions on the thermal density matrix. Therefore, the simulations are computationally expensive due to the fermion sign problem [9], but exact within the given Monte Carlo error bars. We have used $P \sim 10^2$ primitive imaginary-time propagators, which is fully sufficient to ensure convergence at these parameters. Additional details on the simulation of the harmonically perturbed electron gas at finite temperature can be found in Refs. [4].

## C   KS-DFT results for the many-fermion Bohm potential

**The Role of Correlations and Thermal Excitations**. In the manuscript we presented data for interacting electrons at finite temperature at $r_s = 2$ and $\theta = 1$. Although it is clear that the shape of the Bohm potential is determined by both correlations and thermal excitations, a natural question arises: Does the failure of the standard Bohm potential solely result from the effect of electronic correlations or from effects due to thermal excitations? The answer to this question is that the standard Bohm potential does not adequately describe the *many-fermion quantum Bohm potential* when the perturbations are strong regardless of the degree of correlations and thermal excitations. In principle, one can consider various $r_s$ and $\theta$ combinations to show that. However, it is unpractical as the conclusion will always be valid only for the considered $r_s$ and $\theta$ pairs. Instead, we analyze this question as follows: First, we exclude the impact of thermal excitations by considering the zero-temperature limit $\theta \to 0$ (i.e., setting $\theta = 0.01$) and showing that the main conclusion above is valid. Next, we exclude correlation effects by switching off the electron-electron interaction and showing that the main conclusion of the paper regarding the standard Bohm potential holds in this case as well. As a representative case we choose $A = 0.1$.

In Fig. 5, the Bohm potentials for the strongly degenerate case are shown at $\theta = 0.01$. The figure illustrates that the deviation of the standard Bohm potential (dashed blue) from the *many-fermion quantum Bohm potential* (green) is not solely due to thermal excitations.

Then, in Fig. 6, the Bohm potentials for a non-interacting, i.e., ideal electron gas are shown. This figure illustrates that the failure of the standard Bohm potential is not solely an effect of the electron-electron interaction.

Now, as we have established that the deviation of the standard Bohm potential from the correct *many-fermion quantum Bohm potential* is not just due to thermal excitations or the electron-electron interaction, we proceed to analyze the deviations with respect to the wave number of the perturbation.

**Small wave numbers:** $q \lesssim 0.5\, q_F$. For $q = 0.5\, q_F$ we illustrate a comparison of the quantum Bohm potentials (top panel) and their induced forces (bottom panel) in Fig. 7 (where $N = 64$). We observe significant deviations of the standard Bohm potential (dashed blue) from the *many-fermion quantum Bohm potential* (green). The difference in the forces is up to about 50%. In Fig. 8, we further decrease the wavenumber to $q = 0.3\, q_F$ (where $N = 256$). As illustrated, this does not lead to a better agreement, but to larger deviations. However, comparing Fig. 5 with Fig. 8, we notice that with decreasing wave number (from $q \simeq 0.84\, q_F$ to $q \simeq 0.3\, q_F$) also the magnitude of the quantum Bohm potentials decreases from the order of $10^{-2}$ Ha to the order of $10^{-3}$ Ha. This illustrates that the quantum Bohm potential becomes less important in the long wavelength limit as quantum tunneling becomes less important.

**Large wave numbers:** $q > q_F$. Results for larger wave numbers, $q > q_F$, are shown in Fig. 9. In the top panel we illustrate the quantum Bohm potentials for $q \simeq 1.68\, q_F$ and in the bottom panel for $q \simeq 2.53\, q_F$. At $q \simeq 1.68\, q_F$, we observe significant deviations of the stan-

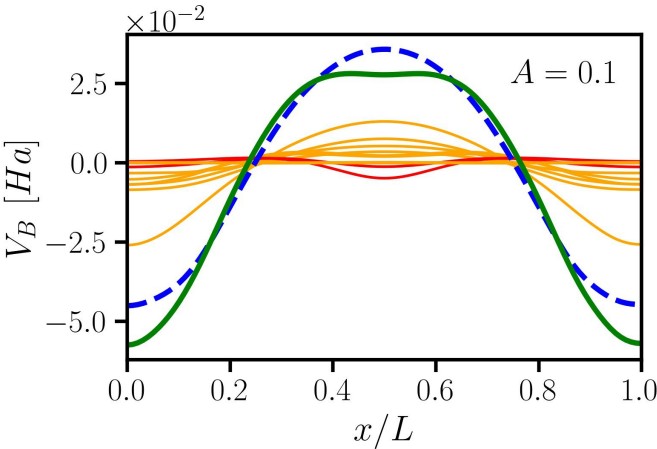

Figure 5: Comparison of the quantum Bohm potentials in the zero-temperature limit at $r_s = 2$ and $\theta = 0.01$. The deviations of the standard Bohm potential (dashed blue) from the exact *many-fermion quantum Bohm potential* (green) are not just due to thermal excitations. The individual contribution of orbitals with one maximum in the center (orange) and with more that one maximum (red) are also illustrated.

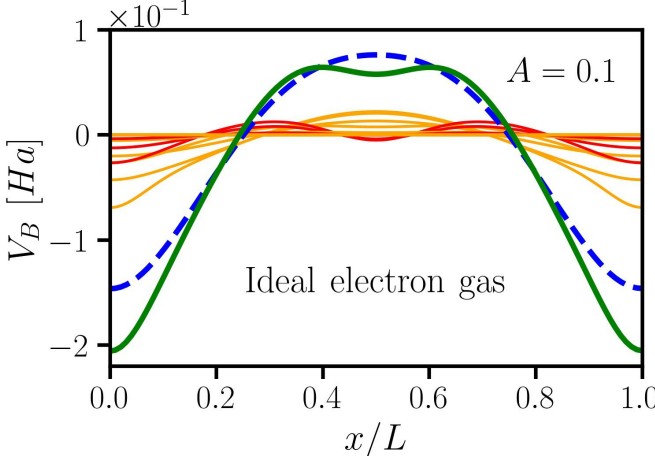

Figure 6: The same as in Fig. 5, but for the non-interacting, i.e., ideal electron gas.

dard Bohm potential from the *many-fermion quantum Bohm potential* in the density depletion regions. As the wave number increases further to $q \simeq 2.53\,q_F$, the deviations decrease. This observation is in agreement with the fact that, in the limit $q \gg q_F$, — which is equivalent to the single-particle limit — the standard Bohm potential becomes exact.

**When is the standard Bohm potential applicable?** As mentioned in the manuscript, the standard Bohm potential is accurate in the limit of weak perturbations. This is demonstrated in Fig. 10, where the quantum Bohm potentials of the correlated electron gas are illustrated for $A = 10^{-3}$. In this case, the maximum deviation in the density from the mean density is about $|\delta n| \simeq 1.8 \times 10^{-3}\,n_0$.

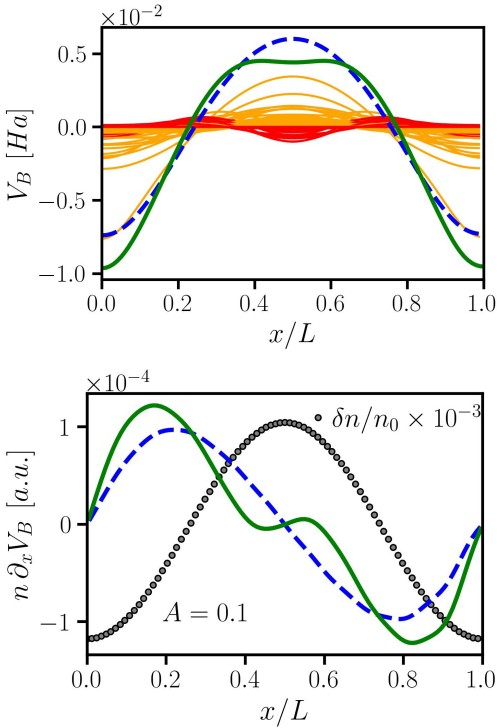

Figure 7: Comparison of the quantum Bohm potentials (top panel) and their induced forces (bottom panel) at $q = 0.5\,q_F$ ($r_s = 2$ and $\theta = 0.01$). The deviations between the standard Bohm potential (dashed blue) and the *many-fermion quantum Bohm potential* (green) persist. The total electron density (grey circles) and the individual contribution of orbitals with one maximum in the center (orange) and with more that one maximum (red) are also illustrated. Note scaling.

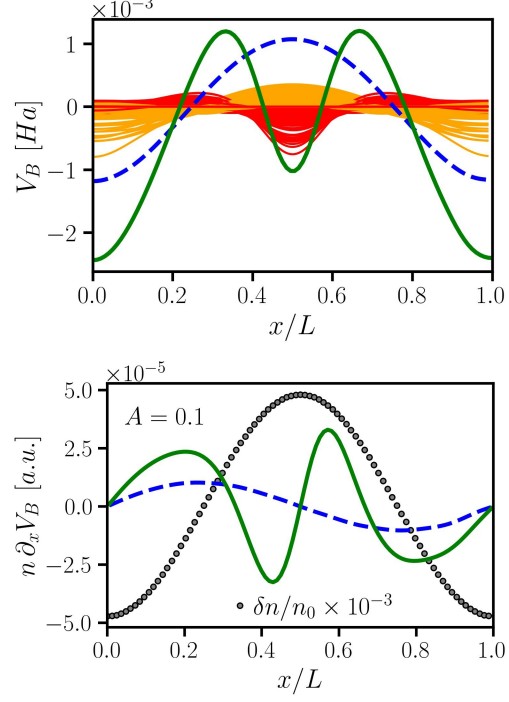

Figure 8: The same as in Fig. 7, but for $q = 0.3\,q_F$.

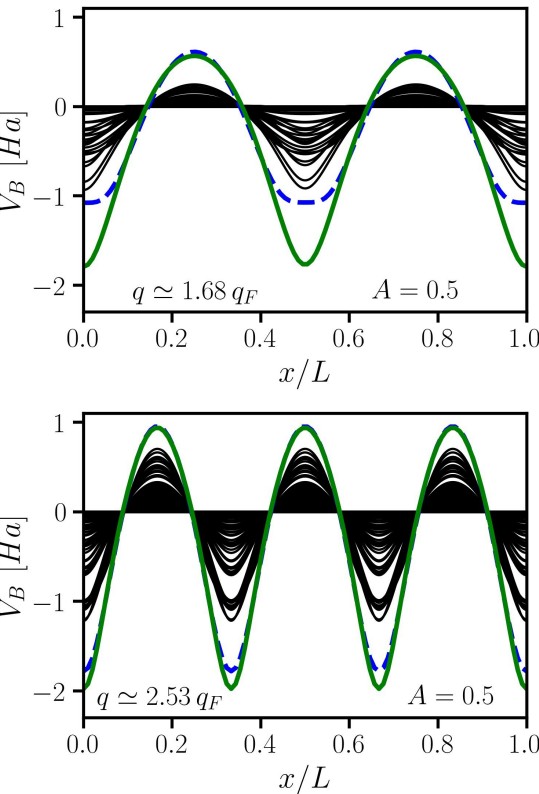

Figure 9: Comparison of the quantum Bohm potentials at $q \simeq 1.68\,q_F$ (top panel) and at $q \simeq 2.53\,q_F$ (bottom panel) ($r_s = 2$ and $\theta = 1$), where we illustrate the standard Bohm potential (dashed blue) and the *many-fermion quantum Bohm potential* (green). We also illustrate the contribution of the orbitals (black) scaled by the factor five (twelve) in the top (bottom) panel.

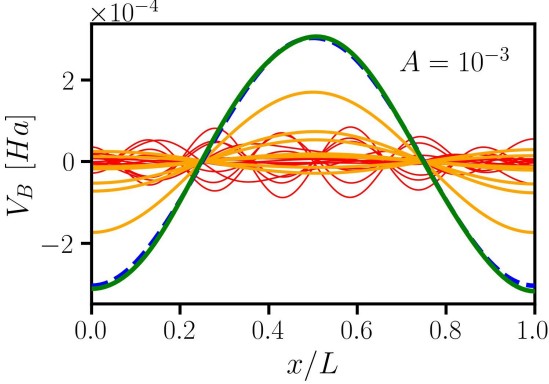

Figure 10: Comparison of quantum Bohm potentials of the correlated electrons gas in the limit of weak perturbation ($\theta = 1$ and $r_s = 2$). The deviations of the standard Bohm potential (dashed blue) from the exact *many-fermion quantum Bohm potential* (green) are shown. The individual contribution of orbitals with one maximum in the center (orange) and with more that one maximum (red) are also illustrated.

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
