# Peer review of "Towards a Quantum Fluid Theory of Correlated Many-Fermion Systems from First Principles"

_SciPost Physics, doi:SciPost Phys. 12, 062 (2022)_

## Round 1 · Referee Report · Anonymous (Referee 1) · 2021-8-1

Report
The present work concerns the investigation of a new approach to the quantum hydrodynamics (QHD) of many-fermion systems. Specifically, the key development consists in considering a many-fermion Bohm potential (MFBP), in contrast to the standard Bohm potential (SBP) that is typically used in current applications.
The appropriateness of the MFBP in the present context is first argued on theoretical grounds, by providing a first-principles derivation starting from the formally exact Kohn-Sham equations. Subsequently, considering a harmonically perturbed electron gas, it is shown that the MFBP significantly differs from the SBP in the presence of large density perturbations. Differences in the resulting forces are of comparable order of magnitude to other terms in the QHD equations, so that significant deviations in the corresponding dynamics can be expected.
The manuscript is interesting and the results presented in it appear sound, in light of the theoretical derivations and the benchmarking provided for the density calculations. Furthermore, the developments of this manuscript appear timely and relevant, due to the great interest in many-fermion systems originating from different fields and the intrinsic limitations of existing numerical approaches, such as Monte Carlo and DMRG. Developments in this direction could have many applications, as argued in the introduction and conclusion.
However, while the results presented in this work are valuable, I believe that the manuscript could benefit from a more definite presentation of what precisely constitutes a new development. For instance, the derivations of the Theory section closely mirror the developments of a previous work of two of the authors, Ref. [11]; see esp. Eq. (37-38) and (42-47) therein. In fact, the MFBP already appears in [11] (although this terminology is not used), where it is argued that the difference between the SBP and MFBP was neglected in previous derivations, and that this can be expected to lead to significant differences when orbital amplitudes are not identical. Furthermore, I think that claims such as “illustrating the failure of the standard Bohm potential” or “this enables more accurate QHD simulations” would be more strongly justified in the presence of an explicit study of the resulting dynamics.
This work’s main advancement is explicitly showing the difference between the MFBP and SBP for a physically relevant scenario, and arguing that this can be expected to significantly affect the resulting dynamics. I believe this to be in itself a relevant result, and I therefore recommend the publication of this manuscript provided the following points are addressed (i.e. either implemented or convincingly rejected):
1. As discussed above, theoretical derivations significantly draw on earlier developments, esp. those of Ref. [11], which should be more thoroughly referenced throughout the Theory section.
2. In order to make the manuscript more self-contained and to facilitate a comparison to earlier methods, it would be beneficial to explicitly provide additional details on how QHD has been performed in earlier works. Specifically: do all mentioned earlier works simply make use of Eqs. (5-6), but with the MFBP being replaced by the SBP? If so, this should be more explicitly stated. If not, any additional differences should be highlighted. (Of course, different approximation methods for the stress tensor etc. could also be used, which are not presently relevant; what would be useful here is to elucidate what exactly is the starting point for earlier approaches vs. the present work.)
3. When Eq. (1) is given, a few Refs could be included where this particular form was used.
4. When Eq. (7) is introduced, it would also be useful to provide some Refs in which this Hamiltonian was previously considered.
5. When mentioning a “breakdown” of the SBP, it would be appropriate to clarify that this is argued on the grounds of the SBP yielding significantly different forces compared to the theoretically motivated MFBP, rather than based on a direct comparison of the resulting dynamics. In fact, while the expected difference in dynamics is convincingly argued by comparing the resulting forces to other relevant terms, it is not explicitly shown, and I think this point should be made clear.
6. Also, the significance of this discrepancy for strong density perturbations is only explicitly shown for one particular system, whereas the wording of the final part of Section 1 might suggest that this is a more general finding, then illustrated with one example.
7. Indeed, since the ultimate aim of the present work is improving a numerical method for time evolution, the paper would greatly benefit from an explicit benchmarking of the dynamics induced by the SBP vs. the MFBP (even for some special case where at least some results are available for comparison), since at present no examples of dynamics are shown.
In addition, I think the manuscript could potentially benefit from the following minor comments and suggestions:
1. In the abstract, it could be useful to clarify “strong density perturbations”.
2. On p. 1, when discussing QHD, it could be helpful to clarify that the “surge of activities” concerns existing applications of QHD.
3. It would be useful to define all quantities as soon as they are first introduced, e.g. n_i, f_i in Eq. (2). Explicit formulae could also be provided, e.g. the equation for f_i as done in [11].
4. On p. 4, when defining q=n q_{min}, n could be confused with the density; a different letter could be used. It would also be helpful to explicitly indicate what n is set to in the various cases.
5. In the caption of Fig. 2, the meaning of the red vs orange lines could also be briefly explained.
6. In the discussion of Fig. 3 on p. 6, following “We infer that…”, forces are improperly referred to as potentials. I would suggest using a more precise wording (e.g. “many fermion / standard Bohm force”).
7. Since the MFBP and SBP are mentioned frequently in the manuscript, the authors could consider using an abbreviation (e.g. using an acronym as done in this report).
The appropriateness of the MFBP in the present context is first argued on theoretical grounds, by providing a first-principles derivation starting from the formally exact Kohn-Sham equations. Subsequently, considering a harmonically perturbed electron gas, it is shown that the MFBP significantly differs from the SBP in the presence of large density perturbations. Differences in the resulting forces are of comparable order of magnitude to other terms in the QHD equations, so that significant deviations in the corresponding dynamics can be expected.
The manuscript is interesting and the results presented in it appear sound, in light of the theoretical derivations and the benchmarking provided for the density calculations. Furthermore, the developments of this manuscript appear timely and relevant, due to the great interest in many-fermion systems originating from different fields and the intrinsic limitations of existing numerical approaches, such as Monte Carlo and DMRG. Developments in this direction could have many applications, as argued in the introduction and conclusion.
However, while the results presented in this work are valuable, I believe that the manuscript could benefit from a more definite presentation of what precisely constitutes a new development. For instance, the derivations of the Theory section closely mirror the developments of a previous work of two of the authors, Ref. [11]; see esp. Eq. (37-38) and (42-47) therein. In fact, the MFBP already appears in [11] (although this terminology is not used), where it is argued that the difference between the SBP and MFBP was neglected in previous derivations, and that this can be expected to lead to significant differences when orbital amplitudes are not identical. Furthermore, I think that claims such as “illustrating the failure of the standard Bohm potential” or “this enables more accurate QHD simulations” would be more strongly justified in the presence of an explicit study of the resulting dynamics.
This work’s main advancement is explicitly showing the difference between the MFBP and SBP for a physically relevant scenario, and arguing that this can be expected to significantly affect the resulting dynamics. I believe this to be in itself a relevant result, and I therefore recommend the publication of this manuscript provided the following points are addressed (i.e. either implemented or convincingly rejected):
1. As discussed above, theoretical derivations significantly draw on earlier developments, esp. those of Ref. [11], which should be more thoroughly referenced throughout the Theory section.
2. In order to make the manuscript more self-contained and to facilitate a comparison to earlier methods, it would be beneficial to explicitly provide additional details on how QHD has been performed in earlier works. Specifically: do all mentioned earlier works simply make use of Eqs. (5-6), but with the MFBP being replaced by the SBP? If so, this should be more explicitly stated. If not, any additional differences should be highlighted. (Of course, different approximation methods for the stress tensor etc. could also be used, which are not presently relevant; what would be useful here is to elucidate what exactly is the starting point for earlier approaches vs. the present work.)
3. When Eq. (1) is given, a few Refs could be included where this particular form was used.
4. When Eq. (7) is introduced, it would also be useful to provide some Refs in which this Hamiltonian was previously considered.
5. When mentioning a “breakdown” of the SBP, it would be appropriate to clarify that this is argued on the grounds of the SBP yielding significantly different forces compared to the theoretically motivated MFBP, rather than based on a direct comparison of the resulting dynamics. In fact, while the expected difference in dynamics is convincingly argued by comparing the resulting forces to other relevant terms, it is not explicitly shown, and I think this point should be made clear.
6. Also, the significance of this discrepancy for strong density perturbations is only explicitly shown for one particular system, whereas the wording of the final part of Section 1 might suggest that this is a more general finding, then illustrated with one example.
7. Indeed, since the ultimate aim of the present work is improving a numerical method for time evolution, the paper would greatly benefit from an explicit benchmarking of the dynamics induced by the SBP vs. the MFBP (even for some special case where at least some results are available for comparison), since at present no examples of dynamics are shown.
In addition, I think the manuscript could potentially benefit from the following minor comments and suggestions:
1. In the abstract, it could be useful to clarify “strong density perturbations”.
2. On p. 1, when discussing QHD, it could be helpful to clarify that the “surge of activities” concerns existing applications of QHD.
3. It would be useful to define all quantities as soon as they are first introduced, e.g. n_i, f_i in Eq. (2). Explicit formulae could also be provided, e.g. the equation for f_i as done in [11].
4. On p. 4, when defining q=n q_{min}, n could be confused with the density; a different letter could be used. It would also be helpful to explicitly indicate what n is set to in the various cases.
5. In the caption of Fig. 2, the meaning of the red vs orange lines could also be briefly explained.
6. In the discussion of Fig. 3 on p. 6, following “We infer that…”, forces are improperly referred to as potentials. I would suggest using a more precise wording (e.g. “many fermion / standard Bohm force”).
7. Since the MFBP and SBP are mentioned frequently in the manuscript, the authors could consider using an abbreviation (e.g. using an acronym as done in this report).
Requested changes
Please refer to the above report.

---

## Round 1 · Referee Report · Anonymous (Referee 2) · 2021-8-27

Report
This paper deals with the Quantum Hydrodynamics (QHD) approach for many-fermions systems. The main aim of the paper is at clarifying the validity of the Bohm potential. The claim of the authors is that this widely used framework fails for strong perturbations. The authors derive a many-fermion Bohm potential, showing that it yields different results as compared with the Bohm potential.
This paper deals with a difficult problem, as no effective computational method for simulating many-fermion systems out of equilibrium exists, especially for two or higher-dimensional systems. Thus, I believe that the paper deserves to be published in Scipost Physics Core.
My main criticism is the following: As far as I understand in using
eq. (5) and eq. (6) to simulate the out-of-equilibrium dynamics one has to determine or find some suitable approximations for
v_xc and P_e. So it is not clear a priori whether having a many-fermion potential is sufficient to obtain trustable results for the out-of-equilibrium dynamics. I think it would be nice if the authors could comment on that in the manuscript.
This paper deals with a difficult problem, as no effective computational method for simulating many-fermion systems out of equilibrium exists, especially for two or higher-dimensional systems. Thus, I believe that the paper deserves to be published in Scipost Physics Core.
My main criticism is the following: As far as I understand in using
eq. (5) and eq. (6) to simulate the out-of-equilibrium dynamics one has to determine or find some suitable approximations for
v_xc and P_e. So it is not clear a priori whether having a many-fermion potential is sufficient to obtain trustable results for the out-of-equilibrium dynamics. I think it would be nice if the authors could comment on that in the manuscript.

---

## Round 2 · Referee Report · Anonymous (Referee 1) · 2021-11-9

Report

The authors addressed all the points that I raised in my previous report improving the manuscript. I think that the paper can be now published.

---

## Round 2 · Referee Report · Anonymous (Referee 2) · 2021-11-9

Report

The authors have addressed the majority of my points satisfactorily. However, I believe there are still a small number of issues that deserve to be addressed more thoroughly before I can recommend the paper for publication. Namely:

1) The paper convincingly shows that the MFBP produces significantly different forces compared to the SBP. This is well explained e.g. at the start of Section 3.
However, as suggested in my earlier comment 5, the wording of the manuscript should reflect the fact that the authors “claim” that the dynamics will significantly differ for the MFBP compared to the SBP, rather than actually showing it (which would require actually computing and comparing the dynamics in the two cases).
For example, as mentioned in my first report, the wording in the abstract “This enables more accurate QHD simulations beyond its common application domain in the presence of strong perturbations at scales unattainable with first-principles methods” is rather strong, given that no dynamics achieving previously unattainable accuracy has actually been shown.
A similar consideration applies to the wording “In conclusion, the many-fermion Bohm potential leads to a substantially different quantum dynamics” at the end of Section 3.

2) In the conclusion, I find the sentences “Particularly, the effect of the standard Bohm potential has very recently been assessed in hydrodynamics simulations (45). These clearly demonstrate the different dynamics obtained and indicate the importance the many-fermion Bohm potential might have on the dynamics of the shock formation” to be unclear.
Since the new preprint (45) only shows the dynamics computed using the SBP, the authors should explain how this by itself shows that the dynamics obtained from the two potentials differs.
In their reply to my first report, the authors note they “already discussed the expected effects of the MFBP on the shock propagation in WDM”. However, their discussion is based on “the presence of higher-order spatial derivatives of the density”. Such higher-order spatial derivative is already present in the SBP, Eq. (1). So, this by itself does not explain the expected difference.

3) In their reply, the authors argue that the manuscript’s results are expected to be general since any perturbation can be expressed as a linear combination of harmonic perturbations. This is true of the perturbation term of the Hamiltonian. However, the force due to the Bohm potential does not depend linearly on the Hamiltonian. Thus, it is not immediately clear to me that the finding that forces are significantly different for harmonically perturbed Hamiltonians implies that this holds for any perturbation. In any case, I think that the manuscript should explicitly mention any arguments the authors have in favour of the general validity of their results, here shown for a particular example.

4) As mentioned in my first report (comment 6), the wording of the final part of Section 1 could make it sound like the authors first show in general that the SBP and MFBP significantly differ, and subsequently turn to a specific example. I would suggest reformulating this for clarity. For example, saying upfront that they focus on a specific example of physical relevance and then that they “(1) generate an exact […]”, or simply changing “We highlight the practical importance of this result by turning our attention to…” to something along the lines of “Throughout the manuscript, we consider the practically important example of…”.

Requested changes

See report.

  • validity: -
  • significance: -
  • originality: -
  • clarity: -
  • formatting: -
  • grammar: -

Author:  Attila Cangi  on 2021-11-12  [id 1941]

(in reply to Report 1 on 2021-11-09)
Category:
answer to question

We thank the referee for reviewing our manuscript and for providing valuable feedback. We address the individual points in the attached file.

Attachment:

Towards_a_Quantum_Fluid_Theory_of_Correlated_Many-Fermion_Sy_PFS989M.pdf

---

## Round 2 · Author Response

Dear Editors,

we thank both referees for their in-depth review of our manuscript and for providing constructive feeback.
We believe we have addressed all concerns and comments raised by the referees and have revised our manuscript accordingly.

We hope our revised manuscript is now suitable for publication.

Sincerely,
Attila Cangi

%----------------%
Response to Referee 2
%----------------%

>This paper deals with the Quantum Hydrodynamics (QHD) approach for many-fermions systems. The main aim of the paper is at >clarifying the validity of the Bohm potential. The claim of the authors is that this widely used framework fails for strong perturbations. >The authors derive a many-fermion Bohm potential, showing that it yields different results as compared with the Bohm potential.

>This paper deals with a difficult problem, as no effective computational method for simulating many-fermion systems out of >equilibrium exists, especially for two or higher-dimensional systems. Thus, I believe that the paper deserves to be published.}

We thank the referee for reviewing our manuscript and for recommending publication. We address the referee's comment below.

%--------%
Comment 1:

>As far as I understand in using eq. (5) and eq. (6) to simulate the out-of-equilibrium dynamics one has to determine or find some >suitable approximations for vxc and Pe. So it is not clear a priori whether having a many-fermion potential is sufficient to >obtain trustable results for the out-of-equilibrium dynamics. I think it would be nice if the authors could comment on that in the manuscript.

We agree that choosing reliable approximations for the terms vxc and Pe is important.
Recent developments in the construction of exchange-correlation functionals are highly relevant. For example, the parametrization of the interacting electron gas at finite temperature by Groth et al. [Phys. Rev. Lett. 119, 135001 (2017)] would be a suitable extension to improve upon vxc.
The electronic pressure Pe can in principle be evaluated exactly according to its definition in terms of an average over orbital contributions. While this is beyond the scope of this manuscript, it would be an interesting analysis. It would give quantitative insight into the common approximation to Pe in terms of the ideal Fermi gas.

We have revised our manuscript below Eq. (6) accordingly:
Proven approximations to the exchange-correlation energy, i.e., vxc can be employed where recent developments such as the parametrization of the interacting electron gas at finite temperature~(36) provide a solid basis for an accurate inclusion of exchange-correlation effects into the QHD equations.
...
Commonly, the electronic pressure Pe is approximated by the ideal Fermi gas.''
%--------%

%----------------%
Response to Referee 1
%----------------%

>The present work concerns the investigation of a new approach to the quantum hydrodynamics (QHD) of many-fermion systems. >Specifically, the key development consists in considering a many-fermion Bohm potential (MFBP), in contrast to the standard Bohm >potential (SBP) that is typically used in current applications.
>
>The appropriateness of the MFBP in the present context is first argued on theoretical grounds, by providing a first-principles derivation >starting from the formally exact Kohn-Sham equations. Subsequently, considering a harmonically perturbed electron gas, it is shown >that the MFBP significantly differs from the SBP in the presence of large density perturbations. Differences in the resulting forces are of >comparable order of magnitude to other terms in the QHD equations, so that significant deviations in the corresponding dynamics can >be expected.
>
>The manuscript is interesting and the results presented in it appear sound, in light of the theoretical derivations and the benchmarking >provided for the density calculations. Furthermore, the developments of this manuscript appear timely and relevant, due to the great >interest in many-fermion systems originating from different fields and the intrinsic limitations of existing numerical approaches, such >as Monte Carlo and DMRG. Developments in this direction could have many applications, as argued in the introduction and conclusion.
>
>However, while the results presented in this work are valuable, I believe that the manuscript could benefit from a more definite >presentation of what precisely constitutes a new development. For instance, the derivations of the Theory section closely mirror the >developments of a previous work of two of the authors, Ref. [11]; see esp. Eq. (37-38) and (42-47) therein. In fact, the MFBP already >appears in [11] (although this terminology is not used), where it is argued that the difference between the SBP and MFBP was neglected >in previous derivations, and that this can be expected to lead to significant differences when orbital amplitudes are not identical. >Furthermore, I think that claims such as “illustrating the failure of the standard Bohm potential” or “this enables more accurate QHD >simulations” would be more strongly justified in the presence of an explicit study of the resulting dynamics.
>
>This work’s main advancement is explicitly showing the difference between the MFBP and SBP for a physically relevant scenario, and >arguing that this can be expected to significantly affect the resulting dynamics. I believe this to be in itself a relevant result, and I >therefore recommend the publication of this manuscript provided the following points are addressed (i.e. either implemented or >convincingly rejected):

We thank the referee for reviewing our manuscript and for providing valuable feedback. We address the individual points below.

%--------%
Comment 1:
>As discussed above, theoretical derivations significantly draw on earlier developments, esp. those of Ref. [11], which should be more >thoroughly referenced throughout the Theory section.}

We agree with this. In the revised manuscript we now point to Ref. [11] at various places where appropriate.
%--------%

%--------%
Comment 2:
>In order to make the manuscript more self-contained and to facilitate a comparison to earlier methods, it would be beneficial to >explicitly provide additional details on how QHD has been performed in earlier works. Specifically: do all mentioned earlier works simply >make use of Eqs. (5-6), but with the MFBP being replaced by the SBP? If so, this should be more explicitly stated. If not, any additional >differences should be highlighted. (Of course, different approximation methods for the stress tensor etc. could also be used, which are >not presently relevant; what would be useful here is to elucidate what exactly is the starting point for earlier approaches vs. the present >work.)

In earlier works, the momentum equation was used along with the Bohm potential containing the mean density (SBP), Eq. (1). There have been a number of different derivations of this result [see Refs. (17; 24; 25), all cited before Eq. (1)].
The bottom line is that indeed in prior works indeed only Eq. (1) has been used, whereas the primary point of this paper is to employ the many-fermion Bohm potential, Eq. (2).

In response to the referee, we implemented the following change in our manuscript. At the end of Sec. 2 we now point out the relation to the traditional QHD approach:

"We stress that in traditional QHD used in prior works, the many-fermion Bohm potential is approximated by the standard Bohm potential in terms of the mean density as defined in Eq. (1)."
%--------%

%--------%
Comment 3:
>When Eq. (1) is given, a few Refs could be included where this particular form was used.

We agree and added a few relevant references (Refs. [17, 24, 25]) before Eq. (1).
%--------%

%--------%
Comment 4:
>When Eq. (7) is introduced, it would also be useful to provide some Refs in which this Hamiltonian was previously considered.

We agree and added several references above Eq. (7):

[37] S. Moroni, D. M. Ceperley and G. Senatore, Static response from quantum Monte Carlo calculations, Phys. Rev. Lett.69, 1837 (1992), doi:10.1103/PhysRevLett.69.1837.

[38] S. Moroni, D. M. Ceperley and G. Senatore, Static response and local field factor of the electron gas, Phys. Rev. Lett.75, 689 (1995), doi:10.1103/PhysRevLett.75.689.

[39] T. Dornheim, M. Böhme, Z. A. Moldabekov, J. Vorberger and M. Bonitz, Density response of the warm dense electron gas beyond linear response theory: Excitation of harmonics, Phys. Rev. Research 3, 033231 (2021), doi:10.1103/PhysRevResearch.3.033231.
%--------%

%--------%
Comment 5:
>When mentioning a “breakdown” of the SBP, it would be appropriate to clarify that this is argued on the grounds of the SBP yielding >significantly different forces compared to the theoretically motivated MFBP, rather than based on a direct comparison of the resulting >dynamics. In fact, while the expected difference in dynamics is convincingly argued by comparing the resulting forces to other relevant >terms, it is not explicitly shown, and I think this point should be made clear.

We have softened the wording, instead of "breakdown" we refer to our observation as "significant differences".
Nevertheless, we believe, that our analysis in terms of forces should be sufficient to claim that the dynamics using the MFBP will significantly differ from the common approach of using the standard Bohm potential. Preliminary results for actual dynamics of shock propagation taking into account the standard Bohm potential have just become available in a preprint (Ref. 44). These results give a good insight into the expected changes when the MFBP would be used. Our revised manuscript now refers to this recent results. More details are given in our reply to the comment below.
%--------%

%--------%
Comment 6:
>Also, the significance of this discrepancy for strong density perturbations is only explicitly shown for one particular system, whereas the >wording of the final part of Section 1 might suggest that this is a more general finding, then illustrated with one example.

While we have shown results for a given perturbation based on one harmonic in Eq. (7), we believe that the presented results are of general character. The reason is that any perturbation can be expressed as a linear combination of harmonic perturbations.
%--------%

%--------%
Comment 7:
>Indeed, since the ultimate aim of the present work is improving a numerical method for time evolution, the paper would greatly benefit >from an explicit benchmarking of the dynamics induced by the SBP vs. the MFBP (even for some special case where at least some results >are available for comparison), since at present no examples of dynamics are shown.

We agree with the referee that our manuscript lays the basis for solving for the dynamics in terms of the QHD equations. Presently, fully time-dependent numerical solutions that employ the MFBP are absent. This is work in progress. Preliminary results (including some of the authors of this manuscript) on employing the SBP for shock propagation have just become available in a preprint which we now cite.

In the conclusions of our original mansucript, we had already discussed the expected effects of the MFBP on the shock propagation in WDM. Based on the recent results described in the new prepint, we can now make our predictions more specific (see 4th paragraph in the conclusion):

"Particularly, the effect of the standard Bohm potential has very recently been assessed in hydrodynamics simulations (45). These clearly demonstrate the different dynamics obtained and indicate the importance the many-fermion Bohm potential might have on the dynamics of the shock formation."

[45] F. Graziani, Z. Moldabekov, B. Olson and M. Bonitz, Shock physics in warm dense matter – a quantum hydrodynamics perspective (2021), arxiv:2109.09081
%--------%

%--------%
Comment 8:
>In addition, I think the manuscript could potentially benefit from the following minor comments and suggestions:

>1. In the abstract, it could be useful to clarify "strong density perturbations".
>2. On p. 1, when discussing QHD, it could be helpful to clarify that the "surge of activities" concerns existing applications of QHD.
>3. It would be useful to define all quantities as soon as they are first introduced, e.g. ni, fi in Eq. (2). Explicit formulae could also >be provided, e.g. the equation for fi as done in [11].
>4. On p. 4, when defining q=nqmin, n could be confused with the density; a different letter could be used. It would also be >helpful to explicitly indicate what n is set to in the various cases.
>5. In the caption of Fig. 2, the meaning of the red vs orange lines could also be briefly explained.
>6. In the discussion of Fig. 3 on p. 6, following "We infer that…", forces are improperly referred to as potentials. I would suggest using a >more precise wording (e.g. "many fermion / standard Bohm force").
>7. Since the MFBP and SBP are mentioned frequently in the manuscript, the authors could consider using an abbreviation (e.g. using an >acronym as done in this report).

We thank the referee for the thorough reading of our manuscript that resulted in these comments and suggestions. We have revised the paper taking into account all items, except the last. We prefer to avoid using these abbreviations in order to keep our manuscript accessible to a broader audience.
%--------%

---

## Round 2 · List of Changes

%--------%
Revision 1:

We have revised our manuscript below Eq. (6) accordingly:
Proven approximations to the exchange-correlation energy, i.e., vxc can be employed where recent developments such as the parametrization of the interacting electron gas at finite temperature~(36) provide a solid basis for an accurate inclusion of exchange-correlation effects into the QHD equations.
...
Commonly, the electronic pressure Pe is approximated by the ideal Fermi gas.''
%--------%

%--------%
Revision 2:
At the end of Sec. 2 we now point out the relation to the traditional QHD approach:

"We stress that in traditional QHD used in prior works, the many-fermion Bohm potential is approximated by the standard Bohm potential in terms of the mean density as defined in Eq. (1)."
%--------%

%--------%
Revision 3:
We added several references above Eq. (7):

[37] S. Moroni, D. M. Ceperley and G. Senatore, Static response from quantum Monte Carlo calculations, Phys. Rev. Lett.69, 1837 (1992), doi:10.1103/PhysRevLett.69.1837.

[38] S. Moroni, D. M. Ceperley and G. Senatore, Static response and local field factor of the electron gas, Phys. Rev. Lett.75, 689 (1995), doi:10.1103/PhysRevLett.75.689.

[39] T. Dornheim, M. Böhme, Z. A. Moldabekov, J. Vorberger and M. Bonitz, Density response of the warm dense electron gas beyond linear response theory: Excitation of harmonics, Phys. Rev. Research 3, 033231 (2021), doi:10.1103/PhysRevResearch.3.033231.
%--------%

%--------%
Revision 4:
4th paragraph in the conclusion:

"Particularly, the effect of the standard Bohm potential has very recently been assessed in hydrodynamics simulations (45). These clearly demonstrate the different dynamics obtained and indicate the importance the many-fermion Bohm potential might have on the dynamics of the shock formation."

[45] F. Graziani, Z. Moldabekov, B. Olson and M. Bonitz, Shock physics in warm dense matter – a quantum hydrodynamics perspective (2021), arxiv:2109.09081
%--------%

---

## Round 3 · Author Response

Response Letter

System Message: WARNING/2 (<string>, line 2)

Title underline too short.

Response Letter
===========

Dear Editors,

we thank both referees for their constructive review. We have revised the paper by taking into account all comments raised by the referees.

We believe our revised manuscript is now suitable for publication.

Sincerely Attila Cangi

Response to Referees

System Message: WARNING/2 (<string>, line 15)

Title underline too short.

Response to Referees
===============

Referee 1

System Message: WARNING/2 (<string>, line 18)

Title underline too short.

Referee 1
=======

Comment 1

System Message: WARNING/2 (<string>, line 21)

Title underline too short.

Comment  1
========

>The authors addressed all the points that I raised in my previous report improving the manuscript. I think that the paper can be now >published.

We thank the referee for reviewing our manuscript and for recommending publication.

Referee 2

System Message: WARNING/2 (<string>, line 29)

Title underline too short.

Referee 2
=======

Comment 1

System Message: WARNING/2 (<string>, line 32)

Title underline too short.

Comment  1
========

>The authors have addressed the majority of my points satisfactorily. However, I believe there are still a small number of issues that >deserve to be addressed more thoroughly before I can recommend the paper for publication.}

We thank the referee for reviewing our manuscript and for providing valuable feedback. We address the individual points below.

Comment 2

System Message: WARNING/2 (<string>, line 39)

Title underline too short.

Comment 2
========

>The paper convincingly shows that the MFBP produces significantly different forces compared to the SBP. This is well explained e.g. at >the start of Section 3. >However, as suggested in my earlier comment 5, the wording of the manuscript should reflect the fact that the authors “claim” that the >dynamics will significantly differ for the MFBP compared to the SBP, rather than actually showing it (which would require actually >computing and comparing the dynamics in the two cases). >For example, as mentioned in my first report, the wording in the abstract “This enables more accurate QHD simulations beyond its >common application domain in the presence of strong perturbations at scales unattainable with first-principles methods” is rather >strong, given that no dynamics achieving previously unattainable accuracy has actually been shown. >A similar consideration applies to the wording “In conclusion, the many-fermion Bohm potential leads to a substantially different >quantum dynamics” at the end of Section 3.}

We agree with these suggestions and have changed the wording in the revised manuscript.

In the revised manuscript, the sentence in the abstract now reads: “This may lead to more accurate QHD simulations beyond its common application domain in the presence of strong perturbations at scales unattainable with first-principles methods.”

In the revised manuscript, the sentence at the end of Section 3 now reads: “In conclusion, the many-fermion Bohm potential may lead to a substantially different quantum dynamics which will be explored in our future work.”

Comment 3

System Message: WARNING/2 (<string>, line 60)

Title underline too short.

Comment 3
========

>In the conclusion, I find the sentences “Particularly, the effect of the standard Bohm potential has very recently been assessed in >hydrodynamics simulations (45). These clearly demonstrate the different dynamics obtained and indicate the importance the many- >fermion Bohm potential might have on the dynamics of the shock formation” to be unclear. >Since the new preprint (45) only shows the dynamics computed using the SBP, the authors should explain how this by itself shows that >the dynamics obtained from the two potentials differs. >In their reply to my first report, the authors note they “already discussed the expected effects of the MFBP on the shock propagation in >WDM”. However, their discussion is based on “the presence of higher-order spatial derivatives of the density”. Such higher-order >spatial derivative is already present in the SBP, Eq. (1). So, this by itself does not explain the expected difference.}

We agree that our statement was unclear. In Ref. (45), the significance of quantum diffraction effects in hydrodynamics simulations provided by the common SBP are demonstrated for shock propagation in WDM. As we show in Figure 2, the magnitudes of the SBP and the many-fermion Bohm potential differ significantly. In particular, the use of the many-fermion Bohm potential leads to stronger forces in region of strong density perturbations (Figure 3). We, therefore, argue in the Conclusion of our manuscript that the shock propagation in WDM is further influenced when the many-fermion Bohm potential is used in hydrodynamics simulations.

We have revised our manuscript to clarify this point [see page 7, end of third paragraph]: "As demonstrated in Figures 2 and 3, the many-fermion Bohm potential yields significantly different forces in regions of strong density perturbation than the standard Bohm potential. We therefore expect the many-fermion Bohm potential to further impact the dynamics of shock formation."

Comment 4

System Message: WARNING/2 (<string>, line 77)

Title underline too short.

Comment 4
========

>In their reply, the authors argue that the manuscript’s results are expected to be general since any perturbation can be expressed as a >linear combination of harmonic perturbations. This is true of the perturbation term of the Hamiltonian. However, the force due to the >Bohm potential does not depend linearly on the Hamiltonian. Thus, it is not immediately clear to me that the finding that forces are >significantly different for harmonically perturbed Hamiltonians implies that this holds for any perturbation. In any case, I think that the >manuscript should explicitly mention any arguments the authors have in favour of the general validity of their results, here shown for a >particular example.

We have softened our language regarding this point. We now explicitly mention that the presented findings are obtained by considering harmonic perturbations [see conclusion, second paragraph]: “Considering a harmonic perturbation in the Hamiltonian defined by Eq.~(7), we showed that the standard Bohm potential...”

Comment 5

System Message: WARNING/2 (<string>, line 89)

Title underline too short.

Comment 5
========

>As mentioned in my first report (comment 6), the wording of the final part of Section 1 could make it sound like the authors first show in >general that the SBP and MFBP significantly differ, and subsequently turn to a specific example. I would suggest reformulating this for >clarity. For example, saying upfront that they focus on a specific example of physical relevance and then that they “(1) generate an exact >[…]”, or simply changing “We highlight the practical importance of this result by turning our attention to…” to something along the >lines of “Throughout the manuscript, we consider the practically important example of…”.

We take up the referee's suggestion and revise the text passage in question accordingly: "Throughout the manuscript, we consider the practically important example of the harmonically perturbed, interacting electron gas at finite temperature which is a challenging many-fermion system and is a relevant for modeling high-energy density experiments conducted at coherent light sources and pulsed power facilities around the globe."

We believe, this now should clearly communicate that the results presented in our manuscript are based on the harmonically perturbed, interacting electron gas at finite temperature.

---

## Round 3 · List of Changes

Last sentence of abstract:
==================
“This may lead to more accurate QHD simulations beyond its common application domain in the presence of strong perturbations at scales unattainable with first-principles methods.”

Page 2, end of second paragraph:
=======================
"Throughout the manuscript, we consider the practically important example of the harmonically perturbed, interacting electron gas at finite temperature which is a challenging many-fermion system and is a relevant for modeling high-energy density experiments conducted at coherent light sources and pulsed power facilities around the globe."

Page 6, end of Section 3:
===========
“In conclusion, the many-fermion Bohm potential may lead to a substantially different quantum dynamics which will be explored in our future work.”

Page 7, second paragraph:
=====================
“Considering a harmonic perturbation in the Hamiltonian defined by Eq.~(7), we showed that the standard Bohm potential...”

Page 7, end of third paragraph:
=====================
"As demonstrated in Figures 2 and 3, the many-fermion Bohm potential yields significantly different forces in regions of strong density perturbation than the standard Bohm potential. We therefore expect the many-fermion Bohm potential to further impact the dynamics of shock formation."

---

## Editorial Decision

published